# Sediment history mirrors Pleistocene aridification in the Gobi Desert (Ejina Basin, NW China)

Georg Schwamborn[1,2,3], Kai Hartmann[1], Bernd Wünnemann[1,4], Wolfgang Rösler[5], Annette Wefer-Roehl[6], Jörg Pross[7], Marlen Schlöffel[8], Franziska Kobe[9], Pavel E. Tarasov[9], Melissa A. Berke[10], Bernhard Diekmann[2]

[1]Freie Universität Berlin, Applied Physical Geography, 12249 Berlin, Germany

[2]Alfred Wegener Institute, Helmholtz Centre for Polar and Marine Research, 14473 Potsdam, Germany

[3]Eurasia Institute of Earth Sciences, Istanbul Technical University, Maslak 34469, Istanbul, Turkey

[4]East China Normal University, State Key Laboratory of Estuarine and Coastal Research, Shanghai 200241, China

[5]University of Tuebingen, Department of Geosciences, 72074 Tuebingen, Germany

[6]Senckenberg Gesellschaft für Naturforschung, 60325 Frankfurt, Germany

[7]Institute of Earth Sciences, Heidelberg University, 69120 Heidelberg, Germany

[8]Institute of Geography, University of Osnabrück, 49074 Osnabrück, Germany

[9]Institute of Geological Sciences, Freie Universität Berlin, 12249 Berlin, Germany

[10]University of Notre Dame, Department of Civil and Environmental Engineering and Earth Sciences, Notre Dame IN 46556, USA

*Correspondence to*: Georg Schwamborn (georg.schwamborn@fu-berlin.de)

**Abstract.** Central Asia is a large-scale source of dust transport, but also holds a prominent changing hydrological system during the Quaternary. A 223-m-long sediment core (GN200) was recovered from the Ejina Basin (synonymously Gaxun Nur Basin) in NW China to reconstructing the main modes of water availability in the area during the Quaternary. The core has been drilled from the Heihe alluvial fan, one of the world's largest continental alluvial fans, which covers a part of the Gobi Desert. Grain-size distributions supported by endmember modelling analyses, geochemical-mineralogical compositions (based on XRF and XRD measurements), and bioindicator data (ostracods, gastropods, pollen and non-pollen palynomorphs, *n*-alkanes with leaf-wax δD,) are used to infer the main transport processes and related environmental changes during the Pleistocene. Magnetostratigraphy supported by radionuclide dating provides the age model. Grain-size endmembers indicate that lake, playa (sheetflood), fluvial, and aeolian dynamics are the major factors influencing sedimentation in the Ejina Basin. Core GN200 reached the Pre-Quaternary quartz- and plagioclase-rich 'Red Clay Formation' and reworked material derived from it in the core bottom. This part is overlain by silt-dominated sediments between 217 and 110 m core depth, which represent a period of lacustrine and playa-lacustrine sedimentation that presumably formed within an endorheic basin. The upper core half between 110 and 0 m is composed of mainly silty to sandy sediments derived from the Heihe River that

have accumulated in a giant sediment fan until modern time. Apart from the transition from a siltier to a sandier environment with frequent switches between sediment types upcore, the clay mineral fraction is indicative for different environments. Mixed layer clay minerals (chlorite/smectite) are increased in the basal Red Clay and reworked sediments, smectite is indicative for lacustrine-playa deposits, and increased chlorite content is characteristic of the Heihe river deposits. The sediment succession in core GN200 based on the detrital proxy interpretation demonstrates that lake-playa sedimentation in the Ejina Basin has been disrupted likely due to tectonic events in the southern part of the catchment around 1 Ma BP. At this time Heihe river broke through from the Hexi Corridor through the Heli Shan ridge into the northern Ejina Basin. This initiated the alluvial fan progradation into the Ejina Basin. Presently the sediment bulge repels the diminishing lacustrine environment further north. In this sense, the uplift of the hinterland served as a tipping element that triggered landscape transformation in the Northern Tibetan foreland (i.e., the Hexi Corridor) and further on in the adjacent northern intracontinental Ejina Basin. The onset of alluvial fan formation coincides with increased sedimentation rates on the Chinese loess plateau, suggesting that the Heihe fluvial/alluvial fan may have served as a prominent upwind sediment source for it.

## 1 Introduction

The aridification of the Asian interior since ~2.95-2.5 Ma (Su et al., 2019) is one of the major palaeoenvironmental events during the Cenozoic. The 'Red Clay' formation and loess deposits on the Chinese Loess Plateau, which are products of the Asian aridification, have been used to broadly constrain the drying history of the Asian interior during the Neogene (Porter, 2007). Studies on these aeolian sequences indicate that aeolian deposits start to accumulate on the Chinese Loess Plateau since ~7-8 Ma (Song et al., 2007), suggesting an initiation of Asian aridification during the late Miocene. Cenozoic uplift of the Tibetan Plateau had a profound effect upon the desertification in the Asian interior enhancing it (Guo et al., 2002). The timing of the uplift of the northern Tibetan Plateau has been under debate for decades and is still so until today, i.e. the onset of intensive exhumation in the Qilian Shan at the north-eastern border of the Tibetan Plateau is thought to occur at ~18-11 Ma and at approximately $7 \pm 2$ Ma (Pang et al., 2019). Wang et al. (2017) suggest an emergence of the Qilian Shan during the late Miocene, the area where the Heihe (engl. = Hei River) evolves from its upper reaches on the northern flanks.

Sediments from the Heihe and the more southeasterly flowing Shiyang Rivers are considered a major source for the Badain Jaran Desert and Tengger Deserts (Yang et al., 2012; Li et al., 2014; Wang et al., 2015; Hu and Yang, 2016). It has been argued that they belong to the dust sources for the Chinese Loess Plateau (Derbyshire et al., 1998; Sun, 2002; Che and Li, 2013; Pan et al., 2016; Yu et al., 2016). Today, the Heihe flows from the Hexi Corridor through the Heli Shan northwards into the Ejina Basin (synonymous: Gaxun Nur Basin), where it forms a giant alluvial fan (Fig. 1). When arriving at the lower reaches of the Heihe, the river carries not only the sediments eroded from the Qilian Shan, but also sediments washed from the western Beishan by ephemeral streams, and silty sands blown in from Mongolia in the North (Li et al., 2011; Che and Li, 2013). In addition, ephemeral channels originating from the eastern Altay Mountains indicate that large amounts of sediments are transported from the Mongolian Altay to the Ejina Basin. During the local wet periods of marine isotope

stages (MIS) 3 and 5, and the mid-Holocene (Yang et al., 2010; 2011), strong fluvial input from the Altay Mountain ranges can be expected (Wünnemann et al., 2007).

The Ejina Basin has a lateral and vertical set of different sediment archives; i.e. lacustrine, playa-lacustrine, aeolian, and fluvial-alluvial (Wünnemann and Hartmann, 2002; Zhu et al., 2015; Yu et al., 2016). Coring the alluvial fan and underlying deposits at a central position within the basin is thus expected to yield a record that constrains the timing and mirrors the complex interactions between (i) Quaternary climate forcing of the Heihe discharge, (ii) a tectonic triggering of sediment pulses from the uplifting Qilian Shan, and (iii) internal sedimentation dynamics as they are characteristic of downstream alluvial fan progradation.

The purpose of this study is to reconstruct the palaeoenvironmental change driven by climate and tectonic history in the area based on a sediment core from a distal position of the Heihe alluvial fan. The sediment is used for generating sedimentological data (i.e., grain size, XRF, XRD) that are augmented by information from selected bioindicators (ostracod and gastropod counts, pollen and non-pollen palynomorphs, *n*-alkane abundances and δD values). Based on this multi-proxy dataset, the transition from more humid to more arid conditions in the Ejina Basin during the past 2.5 Ma years is reconstructed.

The studied core GN200 was drilled 2012 to a depth of 223.7 m, with Quaternary deposits reaching a thickness of 222.6 m. The Quaternary strata are underlain by sediments belonging to the regionally widespread Red Clay formation, a set of alternating reddish aeolian sediments and carbonate-rich dark-reddish paleosols of Neogene (Porter, 2007) or late Cretaceous (Wang et al., 2015) age. The discussion on formation and composition of core GN200 considers the Ejina Basin fill as a prominent source of the Chinese Loess Plateau, because of its position in the upwind area.

## 2 Geographical, tectonic and climatic setting

The Ejina Basin is located in the Gobi Desert and part of the Alashan Plateau. It is an intramontane basin bordered by the Heli Shan in the south, the Beishan to the west, the Badain Jaran Desert to the east, and the Eastern Altay Mountains to the north (Fig. 1). The Ejina Basin has developed as a pull-apart-basin between the Northern Tibetan Uplands (i.e., the Qilian Shan) in the south and the Gobi Altay-Tien Shan mountain chain in the north (Becken et al., 2007). There is predominantly a left-lateral transpression acting on the regional upper crust due to the ongoing India-Eurasia collision (Cunningham et al., 1996). Allen et al. (2017, and references therein) describe that seismicity with earthquake magnitudes M > 7 have affected the Qilian Shan and the Hexi Corridor (see Fig. 1) in historic times. Neotectonic activity at the eastern edge of the Ejina Basin was interpreted based on graben geometry detected within crystalline basement using resistivity measurements (Becken et al., 2007; Hölz et al., 2007). Temporal and spatial patterns of fluvial-alluvial and lacustrine deposition are likely influenced by neotectonic movements; e.g., the western basin margin has a subsidence rate of ca. 0.8-1.1 m/ka (Hartmann et al., 2011), whereas in the north-eastern part of the basin the occurrence of seismites illustrates that seismicity has caused sediment rupture in close vicinity to normal fault lines (Rudersdorf et al., 2017).

From south to north, the elevation ranges between 1300 m and 880 m above sea level (asl). The Heihe main stream entering the Ejina Basin has a length of more than 900 km (Li, X. et al., 2018) and originates from the slopes of the Qilian Shan in the south. From its upper reaches, it flows through the foreland of the Hexi Corridor and arrives at the lower reaches with two branches that are likely controlled by fault lines. Here, the Heihe builds up one of the world's largest continental alluvial fan systems in the endorheic Ejina Basin (Hartmann et al., 2011).

The Heihe basin covers an area of approximately 28,000 km2, while the total catchment of the Heihe system, connected with glaciers in the Qilian Shan (>4000 m a.s.l.), comprises roughly 130,000 km2. Along the distal part of the basin, three terminal lakes, namely Ejina, Sogo Nur and Juyanze, form a chain of lakes, which presently are all dried up (Wünnemann et al., 2007). Radiocarbon dating of ancient shorelines suggests that relative lake-level highstands occurred during MIS 3 (Wünnemann and Hartmann, 2002; Wünnemann et al., 2007b; Hartmann et al., 2011), although the 14C-based chronology for the area may underestimate the timing when compared with IRSL OSL results (Zhang et al., 2006; Wang et al., 2011; Long and Shen, 2015; Li et al., 2018; 2018b).

Presently the winter Siberian Anticyclone dominates the climate conditions in the basin (Chen et al., 2008; Mölg et al., 2013). The study area is characterized by a continental climate that is extremely hot in the summer and cold in the winter; the maximum daily temperature is 41 °C (in July) and the minimum daily temperature is -36 °C (in January). According to data from the Ejina weather station between 1959 and 2015 the mean annual temperature, precipitation, relative humidity, and wind speed were 9.0 °C, 36.6 mm, 33.7 % and 3.3 m/s, respectively, and the mean annual potential evaporation is as high as 3,755 mm (Liu et al., 2016). The growing season in the Ejina Basin is from April to September, during which time it is ice free and has seasonal Heihe river runoff. In contrast, the annual precipitation in the upper reaches in the Qilian Shan reaches 300-500 mm (Wang and Cheng, 1999; Wünnemann et al., 2007). Today, only the Ejina Oasis near Juyanze palaeolake receives ephemeral water input. Typical geomorphological features in the Ejina Basin are gravel plains, yardangs, playas, sand fields, and sporadically distributed mobile linear and barchan dunes (Zhu et al., 2015; Yu et al., 2016).

Modern vegetation of the Alashan Plateau and the foothills of the Qilian Shan is dominated by semi-desert and desert plant communities, mainly consisting of shrubs, dwarf shrubs and low herbs (Herzschuh et al., 2004). *Reaumuria soongorica* (Tamaricaceae) desert has been described in the basin around the coring site (Hou, 2001). The other common desert communities are dominated by *Ephedra przewalskii* and various species of *Artemisia* and Chenopodiaceae (e.g. *Haloxylon*, *Salsola*, *Sympegma*, *Ceratoides*, *Anabasis*), Fabaceae (*Caragana*), *Tamarix* and *Nitraria*. Limited in space steppe vegetation is dominated by various dry resistant grasses (e.g. *Stipa*), shrubs and forbs species. Riparian arboreal vegetation typical for the Hei River banks and former river beds is represented by *Populus euphratica*, *Sophora alopecuroides* and *Tamarix ramosissima* among the dominant taxa (Herzschuh et al., 2004). Sedge (*Carex*), grass (Poaceae) and various forb species are typical members of salty meadow and marshy vegetation communities (Hou, 2001).

## 3 Methods

A rotational drilling system has been used for coring with 3 m long metal tubes 80-120 mm in diameter. The drilling took place in the centre of the Ejina Basin (42°3'12.96''N, 100°54'14.4''E) at 936 m asl. in a maximum distance to known fault lines. Once a core segment was retrieved, it was pressed out immediately, halved, described and photographically documented. Subtracting core gaps and overlaps the length of the core is 223.7 m with a recovery rate of 96%. On average sampling of 2 to 5 cm thick slices was done three times per meter or in accordance to sediment change for studying various sediment properties as described below (SI, supplementary information).

### 3.1 Non-destructive analyses

After core splitting several non-destructive analyses were carried out including visual description, optical line scanning, magnetic susceptibility analyses and XRF element scanning. Magnetic susceptibility measurements at 1 cm resolution were carried out on one core half using a Bartington MS2E sensor, while the other half was scanned with an Avaatech core scanner to determine semi quantitatively element compositions at 1 cm resolution. We applied a Rhodium tube at 150 μA and 175 μA with detector count times of 10s and 15s for elemental analysis at 10 kV (no filter) and 30 kV (Pd-thick filter). Element intensities were obtained by post-processing of the XRF spectra using the Canberra WinAxil software with standard software settings and spectrum-fitmodels. The element intensities depend on the element concentration but also on matrix effects, physical properties, the sample geometry, and hardware settings of the scanner (Tjallingii et al., 2007). We accepted modelled chi square values ($\chi^2$) <2 as a parameter of measured peak intensity curve fitting for the relevant elements.

### 3.2 Grain-size distribution and endmember modelling analysis

Sediment grain-size distributions were determined using a laser diffraction grain size analyser (Malvern Mastersizer 3000). Prior to laser sizing the samples have been removed from organic carbon using H2O2 oxidation on a platform shaker until reaction ceased. The endmember modelling algorithm (EMMA) after Dietze et al. (2012) and modified by Dietze and Dietze (2019) was applied to the grain size data in order to extract meaningful endmember (EM) grain size distributions and to estimate their proportional contribution to the sediments. Results were translated into a core log that illustrates the succession and thickness of EM types. EM modelling analyses are used to address the main sediment types with their associated energy regimes.

### 3.3 Bulk mineralogy

The mineralogical composition of freeze-dried and milled samples was analysed by standard X-ray diffractometry (XRD) using an Empyrean PANalytical goniometer applying CuKα radiation (40 kV, 40 mA) as outlined in Petschick et al. (1996). Samples were scanned from 5° to 65° 2θ in steps of 0.02° 2θ, with a counting time of 4 s per step. The intensity of diffracted radiation was calculated as counts of peak areas using XRD processing software (MacDiff, R. Petschick in 1999). Mineral

inspection focused on quartz, plagioclase and K-feldspar, hornblende, mica, calcite, and dolomite. Accuracy of this semi-quantitative XRD method is estimated to be between 5-10% (Gingele et al., 2001).

### 3.4 Clay mineralogy

The clay fraction (<2 μm) was separated using settling times according to the Atterberg procedure. Clay particles were oriented using negative pressure below membrane filters and they were mounted as an oriented aggregate mount on aluminium stubs with the aid of double-sized adhesive tape. The analyses were run from 2.49° to 32.49° 2θ on a PANalytical diffractometer. Two X-ray diffractograms were performed; one from the air-dried sample, one from the sample after ethylene glycol vapor saturation was completed for 12 hr. Estimation of clay mineral abundances focused on smectite,
mixed-layer smectite/chlorite (10.6 Å), chlorite, and kaolinite (calculated to a sum of 100%) and is based on peak intensities. Clay analyses were made only from silt-dominated samples.

### 3.5 Fossil counts

Counts of fossils, i.e. ostracods and gastropods, have been conducted from 62 samples each comprising 40-85 g of dry weight. The size fractions of >250 μm, 250-125 μm, and 125-63 μm were examined after wet sieving using deionized water.
Encountered shells were determined to at least the genus level. Shell fragments were also registered, but it has been excluded from further discussion due to the assumption that the material points to reworking. To yield meaningful numbers count results have been normalized to 100 g of dry weight.

### 3.6 Pollen analysis and biome reconstruction

For pollen analysis, a total of 62 samples (each representing a 2 cm thick layer) were taken from the layers of clayey silt
within a 113-217.2 m depth interval with a higher potential for sufficiently good pollen preservation. The samples containing 3 to 5 g of sediment were then treated in the pollen laboratory at the Institute of Geological Sciences (FU Berlin) using the dense media separation method as described in Leipe et al. (2019). The laboratory protocol includes successive treatment of sediment sample with 10% HCl, 10% KOH, dense media separation using sodium polytungstate (SPT with a density of 2.1 g/cm$^3$), and acetolysis. In order to estimate pollen concentration (grains per gram), one tablet with a known quantity of exotic
*Lycopodium clavatum* marker spores (Batchnr. 483216) was added to each sample prior to the chemical treatment following Stockmarr (1971). At least 200 terrestrial pollen grains were counted in the samples with a concentration of more than 500 pollen grains per gram and a moderate to good pollen preservation. The percentages of terrestrial pollen taxa refer to the total pollen sum taken as 100%. The percentages of fern spores, aquatic plants and algae refer to the sum of all pollen and spores. Tilia version 1.7.16 software (Grimm, 2011) was used for calculating individual taxa percentages and drawing the diagram.
Interpretation of pollen records from desert regions is challenging due to several limiting factors, including partial poor pollen preservation, long-distance transport of pollen (e.g. pollen from coniferous and birch trees from mountain forests) and re-deposition of pollen from eroded older sediments (Gunin et al., 1999). Modern surface pollen spectra greatly facilitate the

interpretation of fossil records from the arid regions (Tarasov et al., 1998). In this study, we use a published set of 55 recent pollen spectra from the Alashan Plateau and Tsilian Shan (Herzschuh et al., 2004). This representative dataset from the study

region helped to establish relationships between pollen spectra composition and modern vegetation and was successfully used for interpretation of the Holocene pollen record from the 825 cm long sediment core (41.89° N, 101.85° E; 892 m asl.) from Juyanze palaeolake (Herzschuh et al., 2004). Pollen-based biome reconstruction is a quantitative approach, which was first designed and tested using a limited number of key pollen taxa digitized from the 0 and 6 ka pollen spectra from Europe (Prentice et al., 1996). The method has been further adapted for reconstructing the main vegetation types (biomes) present in

northern Eurasia (Tarasov et al., 1998) and in the desert region around the GN200 coring site (Herzschuh et al., 2004). The latter study presents details of the method and the assignment of the terrestrial pollen taxa found in the surface and the Holocene sediment samples from Juyanze core to the respective biomes. In the current study, we apply the same approach and biome-taxa matrix to the fossil pollen data from GN200 core as described in Herzschuh et al. (2004).

### 3.7 Lipid biomarker analysis

Twenty-six samples were used for lipid biomarker analysis. The study focused on determining the concentration and downcore distribution of $n$-alkanes in the samples. The δD values of two $n$-alkanes ($n$C$_{29}$ and $n$C$_{31}$) were also measured. Sediment was freeze-dried and homogenized, and 18 - 42 grams was extracted using a Dionex Accelerated Solvent Extractor 350 with 9:1 dichloromethane:methanol (DCM:MeOH, v:v). The neutral/polar, fatty acids, and phospholipid fatty acid fractions were isolated from the total lipid extract using an aminopropyl column with 2:1 DCM:2-propanol, 4% glacial acetic

acid in ethyl ether, and MeOH, respectively. The neutral/polar fraction containing the $n$-alkanes was further separated using alumina oxide column and 9:1 Hexane:DCM). A final clean up column to further separate the saturated $n$-alkanes was run using hexane and silver nitrate on silica gel column. A Thermo Trace Ultra ISQ gas chromatograph (GC) mass spectrometer (MS) with flame ionization detection (FID) was used to identify and quantify the $n$-alkanes. Samples were injected in splitless mode at 300°C onto a 30 m fused silica column (Agilent J&W DB-5, 0.25 mm ID, 0.25 μm film thickness) with

hydrogen as the carrier gas. Following a minute hold at 80°C, the GC oven temperature ramped to 320°C at a rate of 13°C/min and with a final hold of 20 minutes. The $n$-alkanes were identified by retention times as compared to standard $n$-alkanes mix and also by MS fragmentation patterns. An internal standard, 5α-androstane, was used for compound quantification. The δD values were determined using a Trace 1310 GC coupled to a Finnigan Delta V Plus isotope ratio mass spectrometer (IRMS). Injection conditions and the GC column were identical to measurement on the GC-FID and the oven

program was as follows: 60°C isothermal for one minute, ramp to 320°C at 6°C/min, and a 12 minute hold at 320°C. The H$_3^+$ factor was determined daily and averaged 4.8 ± 0.3 ppm/mV during the analysis. Minimum peak size used was 2500 mV (amplitude 2). Data was normalized to the Vienna Standard Mean Ocean Water (VSMOW) scale using an A6 $n$-alkane standard mix (Arndt Schimmelmann, Indiana University), injected at the beginning, middle, and end of every run for calibration purposes. Squalane with a known isotopic value was co-injected with samples and A6 standard mix to monitor

instrument accuracy and precision, and an in-house $n$-alkane suite was also used to assess instrument conditions. Squalane

deviation from the accepted value <5‰ for all samples and standards analysed. Due to low abundances of other chain lengths, only long chain n-alkanes ($nC_{29}$ and $nC_{31}$) were measured for δD values. Alkane data are expressed as concentration [μg/g dry weight] and using the following equations.

The Average Chain Length (ACL) quantifies the mean homologue length of a suite of n-alkyl compounds. n-Alkane ($nC_{19}$ to
$nC_{33}$) were calculated using equation (1). $C_i$ refers to the peak area and $i$ represents the number of carbons of each individual chain length.

$$ACL = \frac{\sum(i \times C_i)}{\sum C_i} \tag{1}$$

$P_{aq}$ quantifies the relative input of non-emergent macrophytes to emergent macrophytes and terrestrial plants (Ficken et al.,
2000). The proxy is calculated by the ratio of the sum of abundances of mid-chain n-alkanes to the sum of mid- and long-chain n-alkane abundances as shown in equation (2):

$$P_{aq} = \frac{C_{23}+C_{25}}{C_{23}+C_{25}+C_{29}+C_{31}} \tag{2}$$

**3.8 Statistical treatment**

The mineralogical, geochemical, and lipid biomarker data are of compositional nature, which means that they are vectors of non-negative values subjected to a constant-sum constraint (usually 100%). This implies that relevant information is contained in the relative magnitudes and mineralogical and geochemical data analyses can focus on the ratios between components (Aitchison, 1990). In addition, log transformation will reduce the very high values and spread out the small data values and is thus well suited for right-skewed distributions (van den Boogaart and Tolosana-Delgado, 2013). Compared to
the raw data, the log-ratio scatter plots exhibit better sediment discrimination.

Log-ratios can also minimize the problematic issue that element-compositional data from XRF measurements have a poorly constrained geometry (e.g., variable water content, grain size distribution, or density) and nonlinear matrix effects (Tjallingii et al., 2007; Weltje and Tjallingii, 2008). In addition, they provide a convenient way to compare different XRF records even when measured on different instruments in terms of relative chemical variations. Log-ratios of element intensities are
consistent with the statistical theory of compositional data analysis, which allows robust statistical analyses in terms of sediment composition (Weltje et al., 2015).

Prior to PCA (principal component analysis) and k-means cluster analyses, a centred-log ratio (clr) transformation was applied to the data set following Aitchison (1990). This means element ratios were calculated from raw cps values and smoothed with a 5 pt running mean. Thus, cps values were clr transformed (Weltje and Tjallingii, 2008), whereby elements
measured with 10 kV (Al, K, Ca, Ti, Mn, Fe) were calculated separately from 30 kV elements (Rb, Sr, S, Zr, Cr, Zn, Br).

## 3.9 Chronostratigraphy

The chronology of core GN200 is derived from magnetostratigraphy. Consolidated sediment samples were cut out manually as specimens and placed in plastic boxes of 1.8 cm*1.8 cm*1.6 cm. A total of 567 samples were used. Measurements of the natural remanent magnetization (NRM) and stepwise alternating field (AF) demagnetization were performed at the palaeomagnetic laboratory of Tübingen University using a 2G enterprises DC-4 K 755 squid magnetometer system with an in-line 3-axial AF demagnetizer. For data visualization and interpretation, the software package "Remasoft" (Chadima and Hrouda, 2006) was used, applying PCA (Kirschvink, 1980) for determination of palaeomagnetic directions. Because no control of drilling azimuths was available, the analysis and interpretation of palaeomagnetic directions is based solely on inclinations. All specimens were subjected to stepwise AF demagnetization (steps: NRM, 4, 6, 8, 10, 15, 20, 25, 30, 40, 50, 60, 80, and 100 mT). NRM intensities ranged from 0.01 to 4.2 mA/m with a median of 1.6 mA/m. Demagnetization runs usually provided interpretable results before reaching the noise level of the magnetometer. Most samples exhibited one-component-like or two-component-like demagnetization behaviour. AF demagnetization characteristics and thermomagnetic runs proved magnetite as the main magnetic carrier of the characteristic remanent magnetization (ChRM). The resulting polarity sequence is based on 281 ChRM directions, determined by PCA with a minimum of 4 consecutive demagnetization steps and mean angular deviation (MAD) <10° (Fig. 7). A minimum of two subsequent ChRM directions with inclinations >/< +/- 20°, were required for defining a polarity interval. Where necessary, also PCA components with MAD >10° or with demagnetization paths including to the origin (for the final component) were used to support the interpretation. Overprints of recent Earth Magnetic Field (EMF; parallel to normal palaeofield direction), which cannot be separated from the palaeoremanence, lead to better grouping of apparently normal palaeodirections and a more scattered distribution of reverse palaeodirections.

We augment the relative ages of the palaeomagnetic datasets with absolute ages using simple burial dating based on in situ-produced cosmogenic nuclides (e.g., Balco and Rovey II, 2008; Granger, 2014). Five samples from different depths were sieved, different grain sizes cleaned, and prepared according to protocols outlined in Schaller et al. (2016). Chemical preparation of the samples was conducted at the University of Tübingen, Germany. [10]Be/[9]Be and [26]Al/[27]Al ratios were measured at the AMS facility at Cologne, Germany. The age calculation for simple burial dating is based on a MatLab script of Schaller et al. (2016). The decay constants used for 10Be and 26Al are $(4.997 \pm 0.043) \times 10^{-7}$ (Chmeleff et al., 2010; Korschinek et al., 2010) and $(9.830 \pm 0.250) \times 10^{-7}$, respectively (see Norris et al., 1983). We used sea level-high latitude (SLHL) production rates of 3.92, 0.012, and 0.039 atoms/(g(qtz) yr) for nucleonic, stopped muonic, and fast muonic [10]Be production, respectively (Borchers et al., 2016; Braucher et al., 2011). The SLHL production rates for 26Al are 28.54, 0.84, and 0.081 atoms/(g(qtz) yr) for nucleonic, stopped muonic, and fast muonic production, respectively (Borchers et al., 2016; Braucher et al., 2011). These production rates result in a SLHL 26Al/10Be ratio of ~7.4. We then scaled the SLHL production rates to the sample locations of this study based on the online tool of Marrero et al. (2016) using the scaling procedure "SA" from Lifton et al. (2014). Depth scaling of the production rates is based on nucleonic, stopped muonic, and

fast muonic adsorption lengths, which are 157, 1500, and 4320 g/cm2, respectively (Braucher et al., 2011). The density of
2.4 ±0.2 g/cm$^3$ is assumed to be constant over the depth of the core.

Depth-to-age transformation was carried out by linear interpolation between the ground surface (present) and a double-dated core portion between 60 m and 53 m core depth from both techniques, i.e., palaeomagnetic and radionuclide results. Based on lithostratigraphy the portion between 223.7 and 222.6 m core depth is interpreted to belong to the pre-Quaternary (>2.6 Ma) Red Clay formation, a widely studied set of strata that is mainly of aeolian origin and widely spread across NW China
(Kukla, 1987; Porter, 2007; Sun et al., 2010; Shang et al., 2018).

## 4 Results

### 4.1 Sediment stratigraphy

Three main sedimentary units are identified in core GN200 (Fig. 2, supplementary information SI). From bottom to top they are as follows: Unit A (223.7-217.0 m) is dominated by coarse-grained layers (fine- to medium-grained sand) interbedded
with fine-grained sediments (clayey silt). Colours change on a submeter-scale from red and orange in the sandier parts to grey in the silt-rich layers. Sediment change can be both sharp and transitional. Occasionally cm-thick white layers indicate carbonate enrichment in the sandy layers. Unit A includes deposits that are interpreted to belong to the Red Clay formation; i.e. at the core bottom between 223.7 and 222.6 m red sandy clay with angular clasts occurs, which is interpreted a fanglomerate. This subunit has a sharp boundary with the grey (anoxic) medium sand layers overlying them at 222.66 m (see
also: hs.pangaea.de/Images/Cores/Lz/Gaxun_Nur/GN200_images_31-223m.pdf).

Unit B (217.0-110.0 m) has a succession of banked clayey silt with an increasing frequency of intercalated coarser grained layers dominated by very fine sand to fine sand towards the top of the unit. Silt portions can stretch over several meters upcore and turn from grey (217.0-200.0 m) to brown-olive colours (200.0-177.0 m). Remarkably, sequences of clayey silt between 210.0 and 200.0 m show successions of mm-thick laminations of white and grey to orange laminae. Brown to
orange colours appear in the middle to upper part of the unit (177.0-110.0 m). Some layers containing coarse sand to very fine gravel form a top subunit (between 120.0-110.0 m). Counts of macrofossils are overall low, if samples are not barren at all. Ostracod remains can be found occasionally in layers scattering above 196 m core depth and up to the top of the unit. They are admixed to greatest extent (to double-digit numbers) in coarse silty sediments; especially at core depths 181.5 m, 138.0-137.0 m, 128.0-127.0 m, 121.0 m, and at 114.8 m. Gastropod shells are found even more rarely. They are encountered
in fine silt at 177.6 m and 138.7 m core depth and in coarse silt layers at 120.6 m and 113.7 m (fine silt). Remarkably, no fossils occur in the laminated fine silt layers between 210.0 and 200.0 m and between 173.0 and 172.0 m.

Unit C (110.0-0.0 m) is a succession of fine and medium sand layers interbedded with silt banks that decrease in frequency towards the top. The gradational increase in fine silt content in these silt banks is paralleled by a loss of the clay fraction. Depending on dominating grain sizes colours change from yellow, grey and orange in the sandier layers to light-red in the
silt banks. At core depths between 28.5 and 27.5 m and between 10.0 and 7.0 m black soft mud occurs, which likely

represents lake sedimentation. At 103.2 m, at 99.3 m, at 68.6 m, and at 43.2 m core depth noteworthy accumulations of ostracod shells were found. Dominating grain size fractions in layers containing ostracods range from coarse silt to fine and medium sand. Gastropod shells were found only in few layers at 103.2 m, 102.9 m, and 99.9 m depth. The dominating grain-size fractions in these sediments range from coarse silt to fine and medium sand.

A graphic log of the sediment column derived from visual logging and EMMA-derived endmember calculations is presented in Fig. 2. For completion the sample population and EM modelling results are added. Most of the GN200 samples have a polymodal grain size distribution. The EMMA algorithm produces a five-EM model that envelops all main modes. This explains more than 99.6% of the total variance (see App.). EM 5 is associated with medium to fine sand and a primary mode at ~400 μm and a subordinate mode at 37 μm (8.6%); EM 4 represents fine sand with the main mode at 180 μm (34%); EM 325 3 is composed of silty sand with the main mode at 92 μm (11%); EM 2 is composed of medium silt with a primary mode at 18 μm and a subordinate mode at 360 μm (27%). Remarkably, EM 2 is similar as EM 5 but with reverse order of mode precedence. EM 1 is composed of clayey silt with a main mode at 4 μm and a subordinate fine sand fraction admixed (19%). Surficial processes and landforms producing sediments as found in GN200 are typically fluvial, fluvial-aeolian, levee and overbank deposition, sheetflood and surface wash (playa) and lacustrine processes (e.g. Zhu et al., 2015). Fig. 2 illustrates 330 the frequency and depositional interpretation of sediment types in core GN200. Apart from unit A the frequency of coarse-grained (sand-dominated) layers generally increases from the bottom of unit B to the top of unit A. The appearance of the first prominent sand layer that is several meters thick in size is used to set the boundary between the two units B and A.

**4.2 Dating from palaeomagnetism and radionuclide concentrations**

The polarity sequence from core GN200 starts at the top with normal polarity and has several longer intervals of reverse 335 polarity further downcore between c. 60-80 m, 100-135 m, and 170-225 m (Fig. 3). Given that during the Brunhes chron only few very short events of reverse polarity have occurred (Singer, 2014; Cohen and Gibbard, 2019), which cannot explain any longer intervals of reverse polarity, the polarity boundary at c. 60 m can be correlated to the Brunhes/Matuyama (B/M) boundary (0.773 Ma). Locating the exact position of the B/M boundary in the record, however, may be a matter of discussion. The EMF behaviour at the B/M boundary is obviously rather complex (Singer, 2014) and the limitations of the 340 sampling (no azimuths) and demagnetization procedures (no thermal demagnetization possible) do not allow to disentangle the effects of palaeofield behaviour, lock-in-mechanism, and to separate different palaeofield and recent field components completely. In most of the downcore normal polarity intervals, AF demagnetization behaviour of specimens looks one-component-like, whereas in reverse intervals and near reversals it frequently appears more complex and may exhibit two components of magnetization, which cannot be separated sufficiently. From about 55 to 60 m, shallow normal and reverse 345 components with inclinations < 20° can be observed in many samples. Slightly changing the criteria for defining polarity intervals could shift the polarity boundary (B/M boundary) close to 55 m. Based on the polarity pattern and assuming sediment accumulation rates of similar magnitude, the well-defined intervals of normal polarity at about 85-95 m and 150-170 m may be tentatively correlated to the Jaramillo (0.988-1.072 Ma) and Olduvai (1.788-1.945 Ma) subchrons. However,

it is unclear whether two very short intervals of normal polarity at around 115 m and 125 m represent palaeofield behaviour or are artefacts caused by recent field overprints. The lowermost part of the drill core between 172.0 and 222.6 m shows reverse polarity, which is separated by a hiatus from the rest of the polarity sequence.

Burial dating based on in situ-produced cosmogenic nuclides provides three out of six measurements of 10Be and 26Al concentrations, which have produced reliable results; namely GN200 19a, GN200 21a, GN200 34b (Tables 1 and 2). In contrast, the remaining three samples produced signals close to blank and are discarded from interpretation (GN200 68 a, GN200 102a, GN200 102a, Tables 1 and 2). The samples from core depths 19.1 m, 20.3 m, and 53.1 m had quartz portions high enough for robust measurements (Tables 1 and 2). The upper two samples have yielded ages >2 Ma BP, the one at 53.1 m core depth has an age of 0.84 ±0.12 Ma BP. When accepting the lower sample age, the two upper ages are reversals, which can be explained by reworking of old material that has been eroded and transported from the catchment prior to its final deposition in the Ejina Basin (Fig. 4). Given the error bar of the 53.1 m sample (0.84 ±0.12 Ma) the radionuclide age overlaps with the Brunhes-Matuyama boundary (0.773 Ma BP) on the geomagnetic time scale and which itself has an error bar of ±1% at this chron boundary (Singer, 2014). If this geochronological interpretation is true, it would back up the prominent 20 m thick event with negative inclination below 60 m core depth as belonging to the Matuyama chron.

From magnetostratigraphy the first-order depth-to-age relationship produces a mean sedimentation rate of 9 cm/ka during the last 2.58 Ma in the Ejina Basin (Fig. 4). This assumes an overall balanced change of accumulation and erosion across glacial-interglacial cycles in the area. If the radionuclide dating at 53.1 m (0.84 ±0.12 Ma BP) is included, this sedimentation rate slightly decreases to 6 cm/ka in the upper 53 m, whereas the lower 169 m core have a slightly increased sedimentation rate of 10 cm/ka.

## 4.3 Bulk sediment properties

The graphic log of the GN200 sediment column is combined with downcore XRF element distribution in Figure 5. From the bottom to the top elemental relative concentrations show that sandy sediments of unit A are Al-K dominated, whereas fine silty lacustrine deposits at the bottom of unit B are more Mn- and K-Al dominated. Playa deposits in the lower part of unit B can be characterized by Ca-Ti or by Al-K dominated sediments. The upper part of unit B holds an alternation of Mn and Al-K dominated sediments. In unit C sediments change from the Al-K type to greater portions of Ca-Ti dominated sediments. PCA calculations from 10 kV XRF data (i.e., the main siliciclastic components K, Ca, Ti, Mn, and Fe) reveal that Ca and Ti define the first principal component explaining 40.2% of the total variance. The second component is described by Al and Mn explaining 29.3%. Ca likely reflects a combination of carbonate (detrital or authigenic) content and feldspar composition. In this way it can highlight both fine-grained and coarse-grained sediments. The Ca/Ti ratio illustrates that Ti is enriched especially in unit A as also is the case with K. Mn is depleted when compared with the overlying sediments of unit B and unit A. Within unit B Ca most prominently dominates sediment layers between 182 m to 165 m. Further upcore there are individual layers in unit C at 105-102 m, 91 m, and 68 m, where Ca distinctly dominates over Ti.

Remarkably, the variability of K is increased when more sandy sediments are intercalating with playa deposits; this is valid between 175 and 165 m and for most of unit C. Mn excursions are paralleled by enrichment in S and distinct peaks in magnetic susceptibility. This is particularly true for the lacustrine sediments at the bottom of unit B and playa sediments in the middle part of unit B between 168 and 159 m. S is also enriched between 105 and 102 m, in this case without a marked occurrence of Mn and magnetic susceptibility, but paralleled by peaks of Ca.

As for unit B counts of macrofossils are overall low in unit C, if samples are not barren at all. Ostracods are especially found in double-digit numbers at core depths 103.8 m, 68.6 m, 43.4-43.3 m, 38.7 m, 38.1 m 37.6 m 34.9 m, 33.9 m, and 31.9 m. Ostracod communities are dominated by Ilyocypris sp., which prefers fresh- to brackish-water habitats (Mischke, 2001; Yan, 2017). A high abundance of ostracod valves can be an evidence for short transport with a proximate burial (Mischke, 2001). Gastropods are found in considerable double-digit numbers only in sandy to silty sediments between 103.9 m and 99.9 m depth (Fig. 5). They are represented by Radix peregra, which thrives in waters with a salt content of up to 33.5 ‰ (Verbrugge et al., 2012).

Bulk mineralogical composition is characterized by high counts of quartz and feldspar in unit A (Fig. 6). Feldspar is relatively enriched over quartz when compared with units B and C. Dolomite is decreased with respect to calcite in unit A when compared to units B and C. Mineralogical differences between units B and C are less distinct, but best expressed with lower quartz and feldspar amounts in B than in C, where the frequency of sandy layers increases. Hornblende and dolomite do not have distinct trends, but show individual peaks scattered units B and C. They appear to be connected to individual sediment layers; i.e. hornblende at 171.0 m, dolomite at 77.0 m. Calcite is found nearly continuously in unit B with respect to the average of all inspected minerals. Towards the upper 40 m of unit C, which are dominated by a succession of sand layers, the calcite amount decreases.

Smectite in the clay mineral record clearly increases in playa lake sediments of unit B (Fig. 7). In contrast, mixed-layer minerals (i.e. chlorite/smectite) have peak occurrence only in unit C, where kaolinite is low. In unit B and C kaolinite is non-conclusive as is chlorite at a first glance. However, there is an upcore trend towards higher chlorite amounts; between 223.7 and 130.0 m the average is 20%, whereas above 130.0 m core depth it increases to 27%.

## 4.4 Pollen record

Microscopic analysis of the 62 processed samples showed that only 21 samples had sufficiently high pollen concentrations and were suitable for further pollen counting and interpretation of palaeoenvironments. The remaining 41 samples showed the absence or very few pollen grains. The 21 counted samples are scattered over four depth intervals: 160.07-168.46 m, 193.86 m, 197.68-203.41 and 207.3-217.02 m. The identified pollen and non-pollen palynomorphs include 49 terrestrial taxa (trees, shrubs, forbs, herbs, sedges and grasses), 2 taxa representing aquatic plants, 4 types of algae remains as well as fern spores. The main results of pollen analysis and pollen-based biome reconstruction are shown in the summary diagram (Fig. 8).

The core interval 207.3-217.02 m represents the period of time between ca. 2.523 and 2.410 Ma. The analyzed nine samples from this interval show relatively high pollen concentrations, which range from 13,738 to 106,543 grains/g. Among 42 identified taxa, desert taxa such as Chenopodiaceae (62-78%), *Ephedra* (up to 19%), and *Nitraria* (0.5-1.8%) are absolutely dominate pollen assemblages. Arboreal taxa representing the mountain forest include *Pinus* (up to 6.4%), *Picea*, *Abies* and *Betula*, while *Ulmus* (up to 9.6%), *Salix*, *Hippophae* and *Elaeagnus* represent riparian forest communities. Poaceae pollen does not occur regularly and never exceeds 2%. Pollen of *Sparganium* (2-8%) is found in all samples and represent the aquatic shallow-water environments, along with the remains of algae. The fossil pollen spectra composition resembles modern pollen spectra from the Alashan Plateau, collected from a landscape covered with shrubby desert vegetation consisting of Chenopodiaceae, *Nitraria* and *Ephedra* species (Herzschuh et al., 2004). The arboreal pollen from the mountain forests makes a relatively small contribution to the pollen assemblages, which probably reflect a greater-than-present distance to these forests or/and even lesser area occupied by coniferous and birch trees.

The core interval 197.68-203.41 m represents the period of time between ca. 2.363 and 2.294 Ma. Four samples were counted from this interval. The pollen concentration is relatively low (from 3569 to 4720 grains/g) and increases to 12,834 grains/g in the sample 63 (Fig. 8). A total of 39 taxa were determined. The percentages of Chenopodiaceae (19-64%) and *Ephedra* (2.5-15%) pollen decrease towards the top. Among the temperate deciduous tree taxa *Betula* dominate (up to 26%), followed by *Ulmus* (up to 15%) in the two upper samples. *Abies* pollen (4%) is only found in the uppermost sample 61. The samples contain pollen of aquatic taxa such as *Sparganium* (up to 6%) and *Typha latifolia* (up to 10%) and the remains of green algae. The observed changes in the pollen composition suggest a transition from an arid shrubby desert environment similar to the previous interval to a less arid one. Riparian vegetation and taxa of mountain forests are much better represented in the pollen assemblages dated to ca. 2.317-2.294 Ma.

The same trend is observed in the sample 60 from a depth of 193.86 m (Fig. 8), which dates to 2.249 Ma. The pollen concentration is 4720 grains/g. *Ulmus* (16.9%) remains the most visible taxon of the riparian forest, while *Picea* (12.1%), *Betula* (15.5%), and *Pterocarya* (5.8%) represent mountain forest community. *Ephedra* pollen is relatively rare (3.4%) and Chenopodiaceae values are close to minimal (29%) in the entire record. *Typha latifolia* pollen reaches a maximum (11.5%) in this sample. The pollen composition indicates a further decrease in the climate aridity and spread of the temperate zone mountain forest in the upper reaches of the Heihe.

In the interval 160.07-168.46 m, seven samples representing the time period between ca. 1,954 and 1,857 Ma were counted. With the exception of sample 5 (15,979 grains/g), pollen concentrations are low (1449-3584 grains/g). Among the 40 taxa identified, the arboreal taxa are still abundant, including Pinaceae (18%), *Betula* (up to 12%), *Picea* (up to 12%) and *Pinus* (up to 9%). However, Chenopodiaceae (29-54%) and *Ephedra* (up to 17%) increase in abundance. The largest share of algae can be traced to *Botryococcus* (up to 12%). Sample 5 at a depth of 166.92 m, dated to about 1.935 Ma, shows a relatively high pollen concentration and a very high proportion of Chenopodiaceae (76%) and *Ephedra* (11%) and largest number of corroded pollen grains. This suggests that higher pollen concentrations in the record are associated with a greater role for

chenopods (known as very high pollen producers) in regional vegetation, indicating an increase in aridity (Herzschuh et al., 2004; Hou, 2001).

### 4.5 *n*-Alkane abundances and δD record

Evidence for hydroclimate driven vegetation change in the Ejina Basin is provided from biomarker data (Fig. 9). The concentration and distribution of *n*-alkanes allow insight into the vegetation dynamics in the Ejina Basin and its catchment. Plants produce a waxy coating on the surface of their leaves that protect them from desiccation (Eglinton and Hamilton, 1967). These waxes can be transported by wind or water to the sediments where they are robust over geologic timescales (Eglinton and Hamilton, 1967). *n*-Alkanes are often found in the wax of vascular plant leaves and exhibit a strong predominance of odd instead of even number of carbon atoms in a chain (Eglinton and Hamilton, 1967). While long carbon chain lengths are commonly more dominant in terrestrial higher plants ($n$C$_{27}$-$n$C$_{35}$), aquatic algae and microbes are often predominantly composed of shorter chain lengths ($n$C$_{17}$-$n$C$_{21}$) (Ficken et al., 2000). The mid-chain length homologues $n$C$_{23}$-$n$C$_{25}$, often produced in smaller quantities by terrestrial higher plants, are often found in abundance in aquatic macrophytes (Cranwell, 1984; Ficken et al., 2000).

The short-chain $n$C$_{19}$ *n*-alkane is most abundant between core depths 64 m to 44.1 m. Concentrations though are low and range between 0.048 and 0.068 µg/g dry weight of sediment. Mid-chain *n*-alkanes such as $n$C$_{23}$-$n$C$_{25}$ can be found in samples between 217 m to 44.1 m and have higher concentrations with a maximum value of 0.123 µg/g dry weight occurring at 159.2 m depth. Long-chain *n*-alkanes $n$C$_{27}$-$n$C$_{33}$ are dominant in core GN200 with a maximum value of 0.928 µg/g dry weight at 215.45 m core depth. We examine *P*aq, which is a proxy ratio that highlights the terrestrial, emergent aquatic, and submerged aquatic macrophyte origins of the lipids (Ficken et al., 2000). With a few exceptions, much of the record has low *P*aq values around 0.1, indicating that the *n*-alkanes likely originate from terrestrial rather than aquatic sources (Fig. 9).

The δD values of long chain, terrestrially-derived leaf wax *n*-alkanes show a strong linear relationship to precipitation δD values across a wide range of environments (Sachse et al., 2012; Hou et al., 2008; Sachse et al., 2004). In the nearby Quidam Basin, Koutsodendris et al. (2018) interpreted δD of *n*-alkanes as sensitive recorders of palaeoclimatic variability, particularly sensitive indicators of temperature and moisture source variability (Gat, 1996, Sachse et al., 2012, Yao et al., 2013). Here, δD measurements based on leaf-wax $n$C$_{29}$ and $n$C$_{31}$ alkanes yielded δD values between -189‰ and -148‰, and -184‰ and -148‰, respectively. The δD wax values from the $n$C$_{29}$ and the $n$C$_{31}$ alkanes are highly correlated ($r^2 = 0.95$), which suggests a similar origin, thought to be from terrestrial plants.

### 5 Discussion

Earth surface dynamics include a variety of processes that result in mixing of grain size subpopulations in sedimentary systems. Sediment from different sources can be transported and deposited by a multitude of sedimentological processes that have been linked to climate, vegetation, geological and geomorphological dynamics as discussed in Dietze and Dietze

(2019). The record from core GN200 has variable grain size distributions (Fig. 2C) indicating various transport processes that have shaped the depositional environment in the endorheic Ejina Basin. The interpreted endmembers fluvial, aeolian, playa (or sheetflood), and lacustrine processes have smooth transitions reflecting several energy regimes as is typical for desert and alluvial fan environments (Blair and McPherson, 1994). Only recently Yu et al. (2016) described alluvial gravels, fluvial sands, aeolian sand, sandy loess, and lacustrine clays as main sediment types that can be found in the Ejina Basin. Interpretations of nearby cored sediments (230 m long core D100) have related the coarse-grained portions - resembling EM 5 in GN200 - to high-energy fluvial transport from local areas such as the northern (Gobi-Altay-Tienshan range) and western (Beishan) catchment of the basin (Wünnemann et al., 2007b). Following this study well-sorted fine sand as found in core D100 resembles EM 4 in GN200 and is likewise interpreted as being of aeolian to fluvial origin. Coarse and fine silt deposits resemble EM 3 and EM 2 and indicate playa-like depositional environments under different energy systems. Successions of finer to coarser silt layers building up much of unit B suggest that the depositional processes involved alternating hydrological conditions. Possibly this includes occasional desiccation events in the playa plain as is visible from individual layers of well-sorted fine sand, which suggest aeolian deposition at the site. The entire absence or very poor preservation of pollen in 41 out of 62 selected samples taken from a 57-meter section of GN200 core, mainly representing these playa-like sedimentation environments, confirms our interpretation. On the other hand, bioindicators such as ostracods and gastropods document temporarily subaquatic conditions as are found in ponds and playa-lakes. This is also true for unit B sediments, which are interpreted to represent playa-lake environmental conditions.

Formation and transformation of clay minerals in soil profiles and regoliths is determined by an interaction between the geology, drainage control by geomorphology, and the climate of the source terrain (Singer, 1984; Hillier, 1995; Wilson, 1999; Dill, 2017). Tracking clay mineralogical changes in the detrital sedimentary compositions of the Ejina Basin by means of XRD data thus can aid the interpretation of environmental changes. Variations in the Ejina Basin clay mineralogy appear to be closely linked to main changes in depositional environments: mixed-layer clays characterize sediment layers belonging to the Red Clay formation (223.7-222.6 m) and overlying deposits that have incorporated reworked portions of it (222.6-217.0 m) (unit A), smectite-rich clay characterizes the playa environment (large parts of unit B), and chlorite-rich clays are transported with Heihe river sediments (unit C with overlap to unit B).

Complementary detrital and authigenic signals of sediment origin are preserved in the bulk XRD and XRF data and can support the interpretation of sediment environments (e.g. Hillier, 2003; Jeong, 2008; Song et al. 2009). As with the clay signals the unit A sediments, which belong to the Red Clay formation, are well defined by XRD bulk data; i.e., unit A is markedly dominated by quartz and feldspar when compared with unit B and unit C (Fig. 6).

SEM and XRD analyses of samples with higher concentrations of sulphur from nearby core D100 yielded evidence of gypsum formation when sulphur increased (Wünnemann et al., 2007). It has been interpreted as pointing to a stepwise shrinkage of the water body under dry-warm conditions. Within the playa-lake succession in unit B prominent peaks of magnetic susceptibly along with sulphur likely indicate greigite ($Fe_3S_4$) formation (Fig. 5). Preservation of greigite can occur in terrigenous-rich and organic-poor sediments and is proposed to result from a dominance of reactive iron over organic

matter and/or hydrogen sulphide, which otherwise would favour pyritization reactions (Blanchet et al., 2009). In fact, unit B sediments do not contain organic matter based on a set of TOC measurements using an elemental analyser, which produced results only below the detection limit of 0.1% scattered over the unit (not displayed).

The record of *n*-alkanes suggests large glacial and interglacial variability preserved in the record. The discontinuous GN200 biomarker record reveals several intervals where glacial-to-interglacial changes are preserved: between 217 to 210 m a change from glacial to interglacial conditions and between 165 to 148 m a cycle from interglacial to glacial and back to interglacial conditions. Samples further upcore suggest both glacial (128 m, 44 m) and interglacial (64 m, 46 m) conditions. GN200 δD values range between -145‰ (interglacial) and -190‰ (glacial). Considering that the Ejina Basin is located in the mid-latitudes of the Northern Hemisphere and exhibits strong seasonal temperature variability, we interpret leaf wax δD values in GN200 as primarily a measure of temperature and indicative of the origin of moisture following interpretations given in Koutsodendris et al. (2018). As such, on glacial/interglacial timescales, more negative δD wax values should reflect colder rather than wetter conditions and/or a more distant water source. Koutsodendris et al. (2018) further propose that the δD value of $nC_{29}$ and $nC_{31}$ alkanes can be affected by evaporative deuterium enrichment of leaf water caused by enhanced evapotranspiration under low atmospheric humidity based on Sachse et al. (2006), Seki et al. (2011), and Rach et al. (2014). In this way, GN200 samples from dry glacials can be also affected by evapotranspiration.

Discontinuous pollen data from 217-207 m indicate that shrub desert vegetation with a predominance of Chenopodiaceae and *Ephedra* grew close to the coring site and that mountain forests south of the coring site occupied smaller area between 2.523 and 2.410 Ma. This correlates relatively well (within the error of the age model) with the long phase of low precipitation and tundra dominance in the Lake El'gygytgyn record from Northeast Asia (Brigham-Grette et al., 2013; Tarasov et al., 2013). The fact that our pollen record does not reflect changes from glacial to interglacial conditions between 217 to 210 m, as indicated by the biomarker record, may suggest that vegetation and pollen records from the arid region primarily mirror moisture conditions and not the temperature signal, as interpreted here for δD.

In contrast to a former chronology from drilling into the Ejina Basin (i.e., 230 m long core D100, Wünnemann et al., 2007), where a palaeomagnetic dataset has been interpreted to encompass 250 ka BP, the playa-lake environment is now interpreted to extend further back in time. The onset of more humid conditions with lake sedimentation must date back to into the early Pleistocene (>2 Ma BP) based on the Brunhes/Matuyama chron boundary at 60 m core depth and the occurrence of the Jaramillo and Olduvai subchrons interpreted in this study (Figures 3 and 4). A further linear extrapolation of the time axis down to the core bottom is based on the following assumptions: (i) the Gauss chron (normal polarity) is not detected in the record, thus, GN200 reaches the onset of the Matuyama chron at maximum (2.59 Ma). Between 222.6 and 223.7 m a reversal to normal (Gauss?) is indicated, but statistically not significant. But the extrapolated value comes close to the assumed value from linear extrapolation. (ii) The depositional sequence does not change prominently; the core portion between 222.6-172.0 m has an alternation of fluvial-alluvial layers intercalating with playa-lacustrine sediments typical for desert environments. (iii) Unit A sediments (222.6-217.0 m) are interpreted to be reworked material from the underlying Red Clay formation implying that the Neogene likely was at proximity. (iv) Possibly there is a hiatus between the lowermost

layers (223.7-222.6 m) and the overlying sediments (<222.6 m); the core bottom (223.7-222.6 m) consists of fanglomerate sediments and red-coloured medium sandy sediments that are interpreted to represent the Red Clay formation, whereas sediments above a sharp boundary at 222.6 m turn into grey-coloured fine sand to silt layers. The transition from an oxic to an anoxic environment is distinct.

The resulting linear first-order depth-to-age relationship suggests that on a Quaternary time scale the overall sedimentation rates in the Ejina Basin are fairly constant. This matches results from Willenbring and von Blanckenburg (2010), who show that during the Late Cenozoic global erosion rates and weathering are stable; thus demonstrating that erosion and accumulation rates are balanced on Ma-time scales. Even though at smaller scales one may distinguish between independent histories at the subcontinental and basin scales, our age model accepts that extrusion and crustal shortening are complementary processes that have been successively dominant throughout the India-Eurasia collision history (Métivier et al., 1999). This is thought to affect also the Ejina Basin sediment history, which receives detritus from the Qilian Shan in the northern Tibetan Upland. On long time scales (Ma) the sedimentation rates in the Ejina Basin are low, i.e., 9 cm/ka. For comparison, the Tarim and Qaidam basins in the Tibetan Upland received 1 m/ka during the last 2.0 Ma (Métivier et al., 1999).

Other results from surface dating using radionuclides show that the Gobi Desert in the northern margin of the basin developed 420 ka ago, whereas the surfaces that developed from alluvial plains in the Heihe drainage basin formed during the last 190 ka (Lü et al., 2010). The latter developed gradually northward and eastward to the terminal (palaeo) lakes of the river. These temporal and spatial variations in the Gobi Desert are likely a consequence of alluvial processes influenced by Tibetan Plateau uplift and tectonic activities within the Ejina Basin. This largely overlaps with results from Hetzel et al. (2002), who inferred Qillian Shan strike-slip movements from a series of incised terraces dating back to 40 to 170 ka BP using cosmogenic nuclide dating. In addition, Li et al. (1999) suggest that tectonic activity was more intense around 160 ka and 40 ka BP based on 14C and TL dates from dissected alluvial fans in the Hexi Corridor. The relationship between tectonics, surface processes and superimposed climate fluctuations are thus reasonable for at least the past 200 ka. Lü et al. (2010) put forward that possible episodes of Gobi Desert development within the last 420 ka indicate that the advance/retreat of Qilian Shan glaciers during glacial/interglacial cycles might have been the dominant factor to influencing the alluvial intensity and water volume in the basin. Intense floods and large water volumes would mainly occur during the short deglacial periods.

Thus, sedimentologic interpretations of core GN200 have regional palaeoclimatic and palaeotectonic implications. The presence of lacustrine and playa-lacustrine deposits in the Ejina Basin supports previous interpretations of semiarid or arid climatic conditions including indicators such as evaporitic (i.e. sulphur) and possibly greigite bearing deposits in the NW Gobi Desert during the Pleistocene. This climatic interpretation extends previous interpretations to stretch back over a longer time window into the early-to-mid Pleistocene. Former studies presented sediment archives from desert and lake sediments in the area only until MIS 3 (e.g. Hartmann and Wünnemann, 2009; Hartmann et al., 2011) or MIS 5 (Li et al., 2018) or until 250 ka BP (core D100, Wünnemann et al., 2007).

Our pollen data discontinuously covering the core interval between 217 and 160 m (Fig. 9), suggests that the driest phases of the entire record occurred ca. 2.523-2.410 and 2.363-2.338 Ma, the wettest conditions took place ca. 2.317-2.294 and 2.249 Ma, and the phase with intermediate, although unstable, conditions occurred between 1.953 and 1.857 Ma. This is in line with the biome reconstruction, which demonstrates highest scores for the desert biome during the driest (11.6-14.6) and lowest during the wettest (5.7-9.5) phase (Fig. 9). In comparison, the affinity score estimates of the desert biome calculated for the Holocene pollen record from Juyanze palaeolake (Herzschuh et al., 2004) vary between 15 and 19, which indicates an increased role of desert vegetation communities (and greater aridity) during the Holocene interval. A further comparison with the multi-proxy records from Lake El'gygytgyn (Melles et al., 2012; Brigham-Grette et al., 2013) indicates that the reconstructed wet and dry phases in the middle and northern latitudes of East Asia can be broadly synchronous. However, neither the accuracy of the GN200 age model nor the resolution the presented pollen record here allows for more precise conclusions. The same is true when interpreting the *n*-alkane record. When comparing the *n*-alkanes with the desert biome record (Fig. 9) the warm phases indicated by higher δD values (at depths between 217 m to 210 m and between 165 m to 160 m) tend to be mirrored by drier desert biomes.

What effect, if any, did tectonic pulses along the Hexi Corridor to Heli Shan boundary fault have on sedimentation trends in the Ejina Basin? Predictably, tectonic uplift in the source area, i.e. the Qilian Shan (Zheng et al., 2017), should generate periods of regression and/or coarse clastic influx in the adjacent basin, i.e. the Hexi Corridor. It is beyond the scope of this paper to discuss this problem further, but perhaps the cores extracted from the Hexi Corridor (DWJ, XKJD) presented by Pan et al. (2016) act as a support of the possibility of tectonic control on sedimentation in the Ejina Basin. The authors concluded that the Heli Shan opening occurred around 1.1 Ma BP and allowed the Heihe to flow northward into the Ejina Basin (Pan et al., 2016; Fig. 11 therein). The geomorphological change in catchment size, presumably triggered by block movement and/or uplift, would then provide a tipping element that finally led to the expulsion of the distal lake environment in favour of an extensive alluvial fan environment. Alluvial fan progradation may be flanked by changing evaporation rates and humidity changes on glacial-to-interglacial timescale though.

In this sense the first order chronology of the GN200 is confirmed by findings of Pan et al. (2016) and their cores from the Heihe fluvial/alluvial plains in the Hexi Corridor. Thus, the mega-sequence forming unit C in GN200 is a coarsening up succession that represents the arrival and progradation of the Heihe alluvial fan in the Ejina Basin. An upcore enhanced chlorite load, which to some extent is paralleled by an enhanced dolomite load in units B and C, may support this interpretation; chlorite is known to be exposed in basaltic bedrock outcropping in the Qilian Shan and so is dolomite; both minerals are interpreted to be indicative provenance minerals of the southern catchment (Song et al., 2009; Schimpf, 2019), which increase in the Ejina Basin with the arrival of Heihe river sediments.

Pan et al. (2016) discussed a previous study, which suggested that the Shiyang River (400 km SE from Heihe) formed approximately 1.2 Ma (Pan et al., 2007), based on studies of the highest fluvial terraces. This age is consistent with the formation age of the Tengger Desert (Li et al., 2014). A recent study in the Badain Jaran Desert (Wang et al., 2015) suggested a formation age of at least ~1.1 Ma based on electron spin resonance (ESR) dating of aeolian sands from a 310 m

drilling core. First-order dating of the Ejina Basin sediment fill as recovered in GN200 brings additional input into the debate on the timing of when this part of the Gobi Desert started to serve as a sediment source for downwind sediment accumulation such as in the Badain Jaran and Tengger Deserts and ultimately the Chinese Loess Plateau (Chen et al., 2006). The onset of the Ejina alluvial fan formation coincides with increased sedimentation rates on the Chinese loess plateau <1 Ma (Sun and An, 2005; Sun et al., 2010), suggesting that the Heihe sediment fan formation may have served as a prominent upwind sediment source to it. Figure 10 summarizes a depositional model of a progressively northward propagation of the Heihe alluvial fan environment into the Ejina Basin.

The arrival of Heihe sediments coincides with the climate transition during the Mid-Pleistocene. Koutsodendris et al. (2018) discuss that this time, the Mid-Pleistocene Transition (MPT; ~1250-750 ka), is characterized by a change in global climate dynamics associated with the expansion of polar ice sheets (see also Head and Gibbard, 2015; Clark et al., 2006; Raymo et al., 2006). Koutsodendris et al. (2018 and references used therein) discuss further that as a consequence of global cooling during the MPT glaciers formed in high-elevation settings in the low and middle latitudes of both hemispheres. Glacial-interglacial contrasts strengthened after 1200 ka according to Diekmann and Kuhn (2002) based on analysing bulk parameters of a marine sediment core from the southeastern South Atlantic.

Terrestrial records in Central Asia mirror MPT cooling (An et al., 2011; Prokopenko et al., 2006; Sun et al., 2010) and temperature and ice volume change during glacials and interglacials as reviewed in Koutsodendris et al. (2018). For example, core SG-1 from the Qaidam Basin has a record of pollen concentration, $CaCO_3$ content, and magnetic susceptibility that closely tracks global ice volume (Lisiecki and Raymo, 2005) and monsoonal activity in Central Asia (An et al., 2011; Sun et al., 2010) on glacial/interglacial time scales according to Koutsodendris et al. (2018). It has been concluded from a previous sediment core that the Tibetan Plateau may have been glaciated at least to some extent during the MPT based on palynological and $\delta D$ wax-based palaeohydrological data analysis (Koutsodendris et al., 2018). From the same Qaidam Basin record (SG-1) other sediment properties show clear glacial/interglacial humidity changes across the MPT based on magnetic and palynological proxy data (Herb et al., 2013; 2015). Even though GN200 covers the same age range the discontinuous proxy record in concert with the unknown succession of accumulation and erosion hampers a detailed analysis comparable to the palaeolake sediments from the Qaidam Basin. In addition, it is not yet clear, whether GN200 lipid biomarkers have been transported by wind from remote areas or by fluvial input from the catchment or as a mixture of both.

**6 Conclusions**

A 223 m long core (GN200) was drilled in the central part of the Ejina Basin. Multiple parameter analysis of sediment properties illustrates that the basin is filled primarily with playa-lacustrine deposits in the lower half and holds a transition to an increasing frequency of fluvial-alluvial layers in the upper half. The lake environment shrank north-eastwards when from the southwestern part fluvial-alluvial deposits accumulated in a large sediment fan. The Quaternary playa-lacustrine to

fluvial-alluvial deposition is presumably separated by a hiatus from the Neogene/Upper Cretaceous aged Red Clay formation, which is encountered in the bottom six core meters.

    The tipping element that induced the transformation from an early-to-mid Pleistocene more humid and playa dominated environment to a more arid environment dominated by an alluvial fan deposition is likely triggered by uplift and tectonic activity in the upper reaches of the Heihe. This environmental framework is in accordance with the regional environmental

background inferred from other studies. It suggests that the contribution of dust from the Ejina Basin to the Chinese Loess Plateau was relatively limited during the early Quaternary, but may have increased after the progradation of the sediment fan into the basin after about <1 Ma.

## 7 Figure captions

Figure 1. (A) The study site is located in an area dominated by left-lateral transpression due to the ongoing India-Eurasia

collision. GTSFS=Gobi Tien Shan fault system, QSTF=Qilian Shan thrust front, ATF=Altyn Tagh fault. (B) White dotted line: Heihe fan covering much of the Ejina Basin. Black line: Heihe catchment. GN200 marks the coring site. (Service layer credits: Esri, DigitalGlobe, GeoEye, Earthstar Geographics, CNES/Airbus DS, USDA, USGS, AeroGRID, IGN, and the GIS User Community)

Figure 2. (A) GN200 graphic log with sediment units and litho codes deduced from grain size dominating endmembers (EM) (based on Dietze and Dietze, 2019). Interpretation of depositional environment is added. (B) Illustrated endmember calculation results and (C) sample population (see also Appendix). (D) SEM images and core scan examples for the main sediment types as defined by endmember interpretation.

Figure 3. Litho- and magnetotstratigraphy of core GN200. The geomagnetic polarity time scale (GPTS) is from Cohen and Gibbard (2019).

Figure 4. Age-depth-relation in core GN200. The first order age model is related to the interpreted magnetostratigraphy. Radionuclide datings are given in addition. For more discussion see the text.


Figure 5. GN200 core with selected XRF elemental distribution, magnetic susceptibility (SI), and presence of ostracods and gastropods. S clr has been calculated from 30 kV elements. Interpreted greigite occurrence is marked, in addition. (clr = centred-log ratio)

Figure 6. GN200 core with mineral distributions from XRD bulk measurement. Qz=quartz, Fsp=feldspar, Plag=plagioclase, Hb=hornblende, Do=dolomite, Cc=calcite. (clr = centred-log ratio)

Figure 7. GN200 core with clay mineral distribution and interpretative labels. Smec=smectite, ML=mixed layers minerals, Kao=kaolinite, Chl=chlorite. (clr = centred-log ratio)


Figure 8. Percentage pollen diagram summarizing the results of pollen analysis presented in this study. The biome score calculation of the dominant desert biome uses the approach and pollen taxa to biome attribution described in Herzschuh et al. (2004).

Figure 9. GN200 core with concentrations of *n*-alkanes (d.w. = dry weight), average chain length (ACL), coloured areas highlight interpretation of lipid origin (based on Ficken et al., 2000) and δD values with palaeoclimate interpretations versus depth. The desert biome record from Figure 8 is repeated for comparison.

Figure 10. Conceptual model illustrating the progradation of the Heihe alluvial fan into the Ejina Basin. (Service layer
credits: SRTM under CC BY-SA)

## 8 Table list

Table 1: Information for burial ages from drill core GN200

Table 2: Analytical information for burial age calculations

## 9 Appendix

Appendix 1. Default graphical output of robust.EM() as part of the compact protocol, including class- and sample-wise explained variances (top), mean robust loadings as line graphs, mean robust scores as panels of points (bottom). Polygons around loadings and bars around scores represent 1 standard deviation. A legend with main mode position and explained variance of each endmember. Classes span from 0.19-1784 µm. For further reading see Dietze and Dietze (2019).

## 10 Data availability

https://doi.pangaea.de/10.1594/PANGAEA.906582

## 11 Author contribution

GS carried out the sediment sampling and measured various sediment properties. GS also prepared the manuscript with improvements from all co-authors. KH, BW, and BD designed the study. WD carried out the magnetic measurements. AWR carried out the fossil counting and interpretation, MS and FK conducted the pollen analysis, PT wrote the pollen-related text, MAB was in charge of the lipid biomarker measuring program.

## 12 Competing interests

The authors declare that they have no conflict of interest.

## 13 Acknowledgements

This study was funded by the Federal Ministry of Education and Research of Germany (BMBF) as part of the CAME II project (Central Asia: Climatic Tipping Points & Their Consequences), project number 03G0863D and 03G0863E. T. Ehlers, M. Schaller, J. Starke, A. Koutsodendris, and E. Dietze are thanked for helping at various stages of the study. Thanks to the Notre Dame Center for Environmental Science and Technology and Dr. Dana Biasatti, Keith O'Connor, and Alejandra Cartagena Sierra for assistance in the lab.

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
