# Peer review of "Sediment history mirrors Pleistocene aridification in the Gobi Desert (Ejina Basin, NW China)"

_Solid Earth, 2019_

## Referee Comment (RC1) · Attila Ciner (Referee) · 18 Nov 2019

General Comments: This paper is based on a 223 m sediment core obtained from Ejina basin in NW China. The core was drilled on one of the world's largest fan, namely Heihe AF, which constitutes part of the Gobi desert. Several techniques were used such as grain size and end member modeling analyses, geochemical and mineralogical XRF/XRD measurement and bioindicators (fossils, n-alkanes) to understand the palaeoenvironments during the Pleistocene. Cosmogenic burial dating and magentostratigrapy provide the required ages.

The results are clearly indicating that the basin was filled with playa-lacustrine deposits that later pass to fluvial/alluvial systems in time and space. This transition from humid

to much drier conditions (as indicated by dust contribution from the Ejina basin to the Chinese Loess Plateau) is attributed to the uplift of the study area after about <1 Ma.

Technical comments: The English is excellent and is acceptable as is. I just wonder if it is not better to use aeolian with capital A: Aeolian.

Specific comments: Figure 9 shows a section a conceptual model illustrating the progradation of the Heihe AF into the Ejina basin. A map view of this model with different stages would help the reader to better visualise the development of Heihe AF in time and space.

---

## Short Comment (SC1) · 2 Dec 2019

Dear Editors and Authors,

I recently read the discussion paper "Sediment history mirrors Pleistocene aridification in the Gobi Desert (Ejina Basin, NW China)" by Schwamborn et al. with much interest. In doing so, I noticed that many details regarding the organic geochemical analyses are currently missing. The one methods reference cited is a textbook chapter that discusses only in very general terms how biomarkers are analyzed instead of the specific methods information pertinent to this study. Thus, I'd like to suggest the following additions to strengthen the biomarker portion of this manuscript.

1. Leaf wax concentrations: what were the methods used for extraction and column

chromatography? How much sample was extracted? Were additional cleanup steps required to separate saturated and unsaturated alkanes prior to isotopic analysis? Where were the organic geochemical analyses preformed and what make/model instrument was used? Details including type of column, the oven temperature program and flow rates, and the types of internal standards run should be given.

2. Leaf wax deuterium isotopes: The instrument make/model, column type, reactor conditions (temperature, carrier gas and flow rate), and isotopic standards analyzed should be reported. Were the samples analyzed in duplicate or triplicate? What was the minimum peak size used (1000 mV)? How was instrumental error assessed? How often was the H3+ factor determined and what was its range during the analytical period? Was hydrogen isotope drift throughout the life of the reactor evaluated? The recent publications of Goldsmith et al., 2019 JGR Biogeosciences and McFarlin et al., 2019 Quaternary Science Reviews are good examples of the level of detail that should be included when publishing leaf wax $\delta$2H data.

3. In Figure 8, the top two panels should plot the error for the $\delta$2H measurements. For the other panels, it is unusual to present leaf wax concentrations as centered-log ratios. Plotting it in this manner makes it difficult to compare these data with other studies. Usually leaf wax data are presented as concentrations (ng or mg per g sediment extracted) or fractional abundances of the different chain lengths – adding such a plot would be helpful.

I believe that adding this critical information will make these data more readily comparable with other existing leaf wax datasets, which I think will help this work to become better cited in the long-term.

Finally, given that this is an endorheic basin, which should be highly sensitive to changes in aridity, and given that leaf wax $\delta$2H at this location likely reflects both temperature change and shifting moisture sources, has the $\delta$2H difference between a terrestrial compound (nC31) and an aquatic (nC19 or nC21) compound ($\varepsilon$ter-aq) been

examined? This approach has been used by a number of studies to help identify past arid intervals (e.g. Thomas et al., 2018 GRL; Rach et al., 2017; Sachse et al., 2004) and it might work well in an endorheic basin.

Sincerely,

Isla S. Castañeda

---

## Short Comment (SC2) · 3 Dec 2019

Dear editors and authors,

This manuscript presents the results a very interesting project performed in the current desert region of China. A multi-proxy approach has been applied to reconstruct environments of the past 2.5 Myr. Besides the currently presented proxy records, I am aware that also pollen analysis was initially planned and supposedly also performed. However, no pollen results are shown in the submitted version of the paper.

I am wondering why the authors did not include the pollen analysis results, if they are available? These results could be extremely valuable and interesting and may provide a link for correlation with other long-term records, such as those from Lake Baikal or

[Figure]

Lake Elgygytgyn.

Best regards

Christian Leipe

---

## Referee Comment (RC2) · Anonymous Referee #2 · 9 Dec 2019

The paper presents new data from a >200 m core, taken from the Heihe fan in the Ejina Basin. The results are integrated into the regional climatic and tectonic history. Overall, the data are a rich contribution to the regional datasets, and the paper is well worth publishing in Solid Earth. Some edits are needed to improve the paper: Someone needs to go over the English language of the paper and smooth it out. It's far better than I can manage in a second language, but it needs work. Many phrases are too clumsy, e.g. "reconstructing the main transitional modes of water availability..." – from the Abstract. The methods and results sections are very thorough. Figure 1 needs a better image for part B, and a better representation of the major faults in the area. Some poorly-defined faults are sketched, but major and well-understood faults in the Heli Shan and the northern Qilian Shan are left out. Use a better color scale for

the regional topography. Everything looks much the same in the crucial range. Adding a satellite image will help; the fan is beautiful on GoogleEarth, so it is poor that you can't even see it on a figure for the paper. Part A should be replaced by a map closer to the study area – cut out the regions beyond 30-50 N and 80-110 E, and replace by a more detailed map. Figure 1 really is bad. . . Add more description of the lower contact with the Red Clay Formation. As noted in the text, the age of this unit is not well known, with published estimates ranging from Neogene to Cretaceous. But, this study describes intercalation at the base of the core; if correct, this implies a rapid but gradational transition from conditions at the top of the Red Clay to the undoubted Quaternary units. With this relationship, the Red Clay cannot be Cretaceous – at least in the study area. This is an important finding with much more importance than the authors seem to realise. Therefore it is important that they increase the amount of description and discussion of this crucially important part of the section. A change in sediment type and provenance is linked to "opening" of the Heli Shan, about 1 Ma. Again, more description and discussion is needed here. It's as though the authors have latched on to a tectonic argument, but they are not entirely clear what it means. This means that in turn the readers of this paper cannot be sure what is going on. A schematic figure would help. Careful with "tectonic pulses" – these are commonly based on interpretations of the stratigraphy, so if the stratigraphy is used to define a tectonic pulse, and the tectonic pulse is used to explain the change in stratigraphy, there is perfect circular reasoning. I don't think Wang et al (2017) found any evidence for a Pleistocene stepwise uplift in the region – where did this claim come from? The faults on the fan drawn in Rudersdorf et al (2017), reproduced here, are not credible: faults are shown right along the two major streams on the fan surface, there is no evidence for these structures. Once these things are tidied up, the paper will be a very useful contribution to the regional literature.

---

## Author Comment (AC1) · 9 Dec 2019

Following the comments we will (i) expand on the n-alkane section (modify, expand), (ii) include the pollen analysis and interpretation (after late arrival from the lab) and (iii) work on the issues raised by the two reviewers. More co-authors will be included in the course of it.

---

## Author Comment (AC2) · 14 Jan 2020

Thanks go to all reviewers and commentators!
Find below our responses (in yellow) to individual issues raised by the reviewers and commentators:

RC1: 'Sediment history mirrors Pleistocene aridification in the Gobi Desert (Ejina Basin, NW China)', Attila Ciner, 18 Nov 2019

- Figure 9 shows a section a conceptual model illustrating the progradation of the Heihe AF into the Ejina basin. A map view of this model with different stages would help the reader to better visualise the development of Heihe AF in time and space.

We replaced the figure; now there is an oblique map view into the study area including different stages of alluvial fan formation. We hope to meet the reviewer's demand.

RC2: 'Review of Schwamborn et al Ejina Basin paper', Anonymous Referee #2, 09 Dec 2019

- Figure 1 needs a better image for part B, and a better representation of the major faults in the area. Some poorly-defined faults are sketched, but major and well-understood faults in the Heli Shan and the northern Qilian Shan are left out. Use a better color scale for the regional topography. Everything looks much the same in the crucial range. Adding a satellite image will help; the fan is beautiful on GoogleEarth, so it is poor that you can't even see it on a figure for the paper. Part A should be replaced by a map closer to the study area – cut out the regions beyond 30-50 N and 80-110 E, and replace by a more detailed map. Figure 1 really is bad...

We replaced the figure; now we use an optical satellite image. The new figure includes the delineation of the Heihe catchment. However, we removed any fault lines from the map figure, since the article focuses more on sediment properties. In the text we added more details on seismicity of the area (and references). We hope to meet the reviewer's requirements.

- Add more description of the lower contact with the Red Clay Formation. As noted in the text, the age of this unit is not well known, with published estimates ranging from Neogene to Cretaceous. But, this study describes intercalation at the base of the core; if correct, this implies a rapid but gradational transition from conditions at the top of the Red Clay to the undoubted Quaternary units. With this relationship, the Red Clay cannot be Cretaceous – at least in the study area.

We added more lithologic description at due place and refer to imagery of this core part; i.e. *at the core bottom between 223.7 and 222.6 m red sandy clay with angular clasts occurs, which is interpreted a fanglomerate. This subunit has a sharp boundary with the grey (anoxic) medium sand layers overlying them at 222.66 m (see also hs.pangaea.de/Images/Cores/Lz/Gaxun_Nur/GN200_images_31-223m.pdf).*

- A change in sediment type and provenance is linked to "opening" of the Heli Shan, about 1 Ma. Again, more description and discussion is needed here. A schematic figure would help.

We added a statement at due place in the text: *The authors concluded that the Heli Shan opening occurred around 1.1 Ma BP and allowed the Heihe to flow northward into the Ejina Basin (Pan et al., 2016; Fig. 11 therein).*

- I don't think Wang et al (2017) found any evidence for a Pleistocene stepwise uplift in the region – where did this claim come from?

We changed this statement to better reflect the reference. Now: *Wang et al. (2017) suggest an emergence of the Qilian Shan during the late Miocene, the area where the Heihe (engl. = Hei River) evolves from its upper reaches on the northern flanks.*

The faults on the fan drawn in Rudersdorf et al (2017), reproduced here, are not credible: faults are shown right along the two major streams on the fan surface, there is no evidence for these structures.

We deleted the fault lines from the figure.

SC1: 'a few suggestions', Isla Castaneda, 02 Dec 2019

- I noticed that many details regarding the organic geochemical analyses are currently missing. The one methods reference cited is a textbook chapter that discusses only in very general terms how biomarkers are analyzed instead of the specific methods information pertinent to this study.
    - o 1. Leaf wax concentrations: what were the methods used for extraction and column chromatography? How much sample was extracted? Were additional cleanup steps required to separate saturated and unsaturated alkanes prior to isotopic analysis? Where were the organic geochemical analyses preformed and what make/model instrument was used? Details including type of column, the oven temperature program and flow rates, and the types of internal standards run should be given.
    - o 2. Leaf wax deuterium isotopes: The instrument make/model, column type, reactor conditions (temperature, carrier gas and flow rate), and isotopic standards analysed should be reported. Were the samples analyzed in duplicate or triplicate? What was the minimum peak size used (1000 mV)? How was instrumental error assessed? How often was the H3+ factor determined and what was its range during the analytical period? Was hydrogen isotope drift throughout the life of the reactor evaluated? The recent publications of Goldsmith et al., 2019 JGR Biogeosciences and McFarlin et al.,2019 Quaternary Science Reviews are good examples of the level of detail that should be included when publishing leaf wax$\delta$2H data
    - o 3. In Figure 8, the top two panels should plot the error for the$\delta$2H measurements. For the other panels, it is unusual to present leaf wax concentrations as centered-log ratios. Plotting it in this manner makes it difficult to compare these data with other studies. Usually leaf wax data are presented as concentrations (ng or mg per g sediment extracted) or fractional abundances of the different chain lengths – adding such a plot would be helpful.
    - o Finally, given that this is an endorheic basin, which should be highly sensitive to changes in aridity, and given that leaf wax$\delta$2H at this location likely reflects both temperature change and shifting moisture sources, has the$\delta$2H difference between a terrestrial compound (nC31) and an aquatic (nC19 or nC21) compound ($\epsilon$ter-aq) been examined? This approach has been used by a number of studies to help identify past arid intervals (e.g. Thomas et al., 2018 GRL; Rach et al., 2017; Sachse et al., 2004) and it might work well in an endorheic basin.

For covering the n-alkane analysis more appropriate one more co-author has been called (M. Burke).  Chapters 3.6, 4.4, and 5 have been revised to meet much of the comments of SC1. Figure 8 has been modified accordingly.

SC2: 'Suggestion for manuscript improvement', Christian Leipe, 03 Dec 2019

- I am aware that also pollen analysis was initially planned and supposedly also performed. However, no pollen results are shown in the submitted version of the paper. I am wondering why the authors did not include the pollen analysis results, if they are available?

Pollen results arrived late from lab (after first version of the manuscript), but they are indeed available now. We added the new chapters *3.7 Pollen analysis and biome reconstruction* and *4.5 Pollen record,* and more statements in the text at due place (see chapters 2 and 5). In addition, a new figure has been inserted: *Figure 9. Percentage pollen diagram summarizing the results of pollen analysis presented in this study…* For covering the pollen analysis appropriately, more co-authors have been called (M. Schlöffel, F. Kobe, P. E. Tarasov).

---

## Author Response (AR1)

Thanks go to all reviewers and commentators!
Find below our responses (in yellow) to individual issues raised by the reviewers and commentators.

RC1: 'Sediment history mirrors Pleistocene aridification in the Gobi Desert (Ejina Basin, NW China)', Attila Ciner, 18 Nov 2019

- Figure 9 shows a section a conceptual model illustrating the progradation of the Heihe AF into the Ejina basin. A map view of this model with different stages would help the reader to better visualise the development of Heihe AF in time and space.

We replaced the figure (now Figure 10); it is now an oblique map view into the study area with different stages of alluvial fan formation. We hope to meet the reviewer's demand.

RC2: 'Review of Schwamborn et al Ejina Basin paper', Anonymous Referee #2, 09 Dec 2019

- Figure 1 needs a better image for part B, and a better representation of the major faults in the area. Some poorly-defined faults are sketched, but major and well-understood faults in the Heli Shan and the northern Qilian Shan are left out. Use a better color scale for the regional topography. Everything looks much the same in the crucial range. Adding a satellite image will help; the fan is beautiful on GoogleEarth, so it is poor that you can't even see it on a figure for the paper. Part A should be replaced by a map closer to the study area – cut out the regions beyond 30-50 N and 80-110 E, and replace by a more detailed map. Figure 1 really is bad...

We replaced the figure (now Figure 1); we use an optical satellite image. The new figure includes the delineation of the Heihe catchment. However, we removed any fault lines from the map figure, since the article more focuses on sediment core properties.
ln 90-91: In the text we added more details on seismicity of the area (and references). We hope to meet the reviewer's requirements.

- Add more description of the lower contact with the Red Clay Formation. As noted in the text, the age of this unit is not well known, with published estimates ranging from Neogene to Cretaceous. But, this study describes intercalation at the base of the core; if correct, this implies a rapid but gradational transition from conditions at the top of the Red Clay to the undoubted Quaternary units. With this relationship, the Red Clay cannot be Cretaceous – at least in the study area.

ln 300-302: We added more lithologic description at due place and refer to imagery of this core part; i.e. *at the core bottom between 223.7 and 222.6 m red sandy clay with angular clasts occurs, which is interpreted a fanglomerate. This subunit has a sharp boundary with the grey (anoxic) medium sand layers overlying them at 222.66 m (see also hs.pangaea.de/Images/Cores/Lz/Gaxun_Nur/GN200_images_31-223m.pdf).*

- A change in sediment type and provenance is linked to "opening" of the Heli Shan, about 1 Ma. Again, more description and discussion is needed here. A schematic figure would help.

We added a statement:
ln 606-608: *The authors concluded that the Heli Shan opening occurred around 1.1 Ma BP and allowed the Heihe to flow northward into the Ejina Basin (Pan et al., 2016; Fig. 11 therein).*

- I don't think Wang et al (2017) found any evidence for a Pleistocene stepwise uplift in the region – where did this claim come from?

We changed this statement to better reflect the reference. Now:
ln 55: *Wang et al. (2017) suggest an emergence of the Qilian Shan during the late Miocene…*

- The faults on the fan drawn in Rudersdorf et al (2017), reproduced here, are not credible: faults are shown right along the two major streams on the fan surface, there is no evidence for these structures.

We deleted the fault lines from the figure.

SC1: 'a few suggestions', Isla Castaneda, 02 Dec 2019

- I noticed that many details regarding the organic geochemical analyses are currently missing. The one methods reference cited is a textbook chapter that discusses only in very general terms how biomarkers are analyzed instead of the specific methods information pertinent to this study.
  - 1. Leaf wax concentrations: what were the methods used for extraction and column chromatography? How much sample was extracted? Were additional cleanup steps required to separate saturated and unsaturated alkanes prior to isotopic analysis? Where were the organic geochemical analyses preformed and what make/model instrument was used? Details including type of column, the oven temperature program and flow rates, and the types of internal standards run should be given.
  - 2. Leaf wax deuterium isotopes: The instrument make/model, column type, reactor conditions (temperature, carrier gas and flow rate), and isotopic standards analysed should be reported. Were the samples analyzed in duplicate or triplicate? What was the minimum peak size used (1000 mV)? How was instrumental error assessed? How often was the H3+ factor determined and what was its range during the analytical period? Was hydrogen isotope drift throughout the life of the reactor evaluated? The recent publications of Goldsmith et al., 2019 JGR Biogeosciences and McFarlin et al.,2019 Quaternary Science Reviews are good examples of the level of detail that should be included when publishing leaf wax$\delta$2H data
  - 3. In Figure 8, the top two panels should plot the error for the$\delta$2H measurements. For the other panels, it is unusual to present leaf wax concentrations as centered-log ratios. Plotting it in this manner makes it difficult to compare these data with other studies. Usually leaf wax data are presented as concentrations (ng or mg per g sediment extracted) or fractional abundances of the different chain lengths – adding such a plot would be helpful.
  - Finally, given that this is an endorheic basin, which should be highly sensitive to changes in aridity, and given that leaf wax$\delta$2H at this location likely reflects both temperature change and shifting moisture sources, has the$\delta$2H difference between a terrestrial compound (nC31) and an aquatic (nC19 or nC21) compound ($\varepsilon$ter-aq) been examined? This approach has been used by a number of studies to help identify past arid intervals (e.g. Thomas et al., 2018 GRL; Rach et al., 2017; Sachse et al., 2004) and it might work well in an endorheic basin.

For covering the *n*-alkane analysis more appropriate one more co-author has been called; i.e. M.A. Berke. Following comments by SC1 we modified the manuscript as follows:
ln 201-235: chapter 3.6, now chapter 3.7, has been revised,
ln 450-474: parts of chapter 4.4, now chapter 4.5, have been revised,
ln 516-531: parts of chapter 5 have been revised.
Figure 8, now Figure 9, has been modified accordingly.

Some more references have been added to the text and the reference list.
We hope to meet much of the comments raised by SC1.

SC2: 'Suggestion for manuscript improvement', Christian Leipe, 03 Dec 2019

- I am aware that also pollen analysis was initially planned and supposedly also performed. However, no pollen results are shown in the submitted version of the paper. I am wondering why the authors did not include the pollen analysis results, if they are available?

Pollen results arrived late from lab (after first version of the manuscript), but they are indeed available now. For covering the pollen analysis appropriately, more co-authors have been called; i.e. M. Schlöffel, F. Kobe, P. E. Tarasov. Following comments by SC2 we modified the manuscript as follows:
ln 120-128: has been added,
ln 175-200: The new chapter *3.6 Pollen analysis and biome reconstruction* has been introduced,
ln 407-449: The new chapter *4.4 Pollen record* has been introduced,
ln 491-493: has been added,
ln 532-538: has been added,
ln 585-601: has been added.
A new figure has been inserted: Figure 8 Percentage pollen diagram summarizing the results of pollen analysis presented in this study. The biome score calculation of the dominant desert biome uses the approach and pollen taxa to biome attribution described in Herzschuh et al. (2004).
Some more references have been added to the text and the reference list.
We hope to meet much of the comments raised by SC2.

Additional minor corrections have been made. They do not alter main statements or interpretations, but refine existing ones. All changes are marked-up in the manuscript.

[revised manuscript text omitted]

Fig. 3

[Figure]

Fig. 4

[Figure]

Fig. 5

[Figure]

Fig. 6

[Figure]

Fig. 7

[Figure]

Fig. 8

[Figure]

Fig. 9

[Figure]

**0.8-0.0 Ma**       **fluvial-alluvial environment**

GN200

**1.0-0.8 Ma**       **fan progradation**

GN200

**~1.1 Ma**       **onset of fan formation**

GN200

**~2.5-1.1 Ma**       **playa-lake environment**

GN200

A ———————————— A'

**Eastern Altay**

N

**Bei Shan**   Ejina Basin  A'  GN200

Heihe

A

**Hexi Corridor**  **Heli Shan**

**Qilian Shan**

m asl

**Legend**

| | slope and bedrock | | aeolian deposits |
| | silt and clay | | fan and slope deposits |
| | sand and gravel | | playa-lacustrine |

Fig. 10

[Figure]

Appendix 1

---

## Referee Report (RR1)

**Review on the manuscript "Sediment history mirrors Pleistocene aridification in the Gobi Desert (Ejina Basin, NW China)", by Schwamborn et al., journal "Solid Earth".**

**General comments.**
This is a very interesting work dealing with the interactions between the tectonic and climate dynamics through the study of the sediments of a 200-long core extracted from the Ejina Basin in northwestern China. The manuscript is well written and presents a good state of the art / related literature revision. The methods are considered appropriate, well chosen and explained with high detail and wealthy data. The results are considered good, and thoroughly described and explained. And finally, the discussion presents a good interpretation of them, integrating them with the related literature and reaching valuable conclusions for a better understanding of the palaeogeography and palaeoenvironments of the region. The scientific quality is considered very good for the journal, and for the results themselves, and honestly, I only have some minor comments to this work, which consider that deserves being published.

**Abstract.**
Ok, very complete despite their synthetic character, and with many data of interest. It summarizes well the manuscript.

**Introduction.**
This section includes a comprehensive review on the relevance of the study of sediments in this area in relation to the interactions of the tectonics and climate forcing. An appropriate chronological background of the main tectonic events is included together with the dynamics of the sediment transport and accumulation in the area through the main streams, from the origin in the erosion of the main mountain ranges to the formation of the main alluvial fans. With this information, the aim of the paper is well justified. However, I suggest to move the last paragraph (L78-82) to the results section so that the results do no get ahead so soon in the manuscript.

**Study area.**
The information given in this section is very complete and gives a nice and accurate overall picture of the study area /region. In the first part, i.e. geographical setting, I suggest to include some data regarding to the specific location with latitude/longitude. Regarding to the climate, the data provide are adequate, but I wonder if the authors could add some information regarding to the synoptic climatology of the summer (i.e. the winter anticyclone is mentioned but no what happens in summer).

L101/102: change the "2" in "km2" to superscript.
L105: idem for "14" in "14C".
L102: "m/s", better as "m s$^{-1}$". Information about wind direction and seasonal changes susceptible to modify precipitation throughout the year?
L114: please, add a hyphen in "ice free".
L118-126: this comprehensive information about the vegetation may be summarized considering the nature and aims of this paper.

**Methods.**
The methods selected by the authors are considered appropriate for the study, and its use is well justified. The explanation of each method includes very detailed information. However, it seems to me that some sub-sections (i.e. methods/techniques) are explained with more detail and data than others (compare e.g. mineralogy subsections with pollen analysis or chronostratigraphy). Thus, a more balanced explanation of the methods is recommended. Moreover, in most of the cases, due to the use of many abbreviations, symbols and technical information, those readers not very familiarized with these techniques/and methods, may "be lost" when reading through this section. The information given is considered appropriate, but in the case of the sub-section 3.9

some results are given (L260-266), and I suggest to re-locate them and move to the 'Results section'.

L129: what do the authors mean with "maximum distance"? Could they specify it?
L276-277: change the exponent of the power notation and the numbers of the isotopes (mass number in 10Be and 26Al) to superscript.
L278-280: please, correct the units of the production rates to "atoms (g quartz)$^{-1}$ yr$^{-1}$".
L280: "stopped muonic"? Do the authors mean slow muons?
L281: please, format the mass numbers as superscript.
L282: I think that the authors mean an "online calculator" rather than a "tool". But which tool do the authors refer to? CRONUS? Please, clarify it.
L284-285: please format the length and density units to "cm$^{-2}$".
L287: this is a result; thus, please move to the 'Results' section.

**Results.**
The presentation and description of the results is considered good, although possibly, they might be summarized as a slightly shorter extent would benefit the manuscript.

L293: the three units should be mentioned now. Then continue with the description of each one.
L357-359: this is an interpretation, and thus I recommend to move it to the discussion section.
L363, L366, L367: please, change to "cm ka$^{-1}$".
L389: this is an interpretation – please, move it to the discussion section.
L421-422: this is an interpretation – please, move it to the discussion section.
L430-431: this is an interpretation – please, move it to the discussion section.
L437-438: this is an interpretation – please, move it to the discussion section.
L439: be careful with the decimal separators – change to point (decimal) instead of comms (thousands million years).
L445-447: this is an interpretation – please, move it to the discussion section.
L450-458: this information may be summarized, and even relocated: either in the 'Methods' section or in the 'Discussion', due to its usefulness for the interpretation or its relevance itself.

**5. Discussion.**
The interpretation of the results and their explanation in the context of the related literature is very clear and easy to follow. I only have some minor observations:

L526: which is the symbol at the beginning of the line? A typo? "Chenopodiaceae": format it in italics.
L556, L557: please, change "/ka" to "ka$^{-1}$".
L616-618: this reasoning is really interesting. I wonder if the authors could support this finding on some data of regional atmospheric palaeocirculation.
L627: what is "MTP": the meaning of this abbreviation has not been presented before.
L629: "CaCO3": format the "3" as subscript.

**6. Conclusions.**
The conclusions summarize well the findings of the manuscript and the tectonic/climatic palaeogeographic interactions. Nothing else to add. Ok.

**Figures and tables.**
I cannot assess their quality as they are not included together with the manuscript, and wasn't able to see them.

---

## Author Response (AR2)

Thanks got to Referee #3. Find below our responses (underlined) to individual issues raised by the reviewer. All changes made to the text are highlighted with color.

**Introduction**

"I suggest to move the last paragraph (L78-82) to the results section…" Done.

**Study Area**

"I suggest to include some data regarding to the specific location with latitude/longitude." Done; actually, a cut/paste from the "Methods" section.

"… I wonder, if the authors could add some information regarding to the synoptic climatology of the summer…" Done. The following statement has been added (L 110-113): For the seasonal cycle, Liu et al. (2016) have ascertained that there are strong winds, especially in spring and autumn, with maximum wind speeds of 16.5 m/s. West winds prevailing during summer in the area thereby interact with humid air masses of the summer monsoon further south to release occasional heavy rain fall and thunderstorms (Domrös and Peng, 2012), which may occur at least in the Badain Jaran and Tengger Deserts (Wünnemann, 1999).

L101/102: changed to $km^2$

L105: changed to $^{14}C$

L112: Done; changed to $m\ s^{-1}$

L114: Done; hyphen in ice-free.

L118-126: Summary of is demanded. This part has been shortened now: two sentences have been deleted without losing the main message.

**Methods**

L260-266: and I suggest to re-locate them to move to the "Results section". Done.

L129: "maximum distance"? Could they specify it? We deleted the word *maximum*.

L276-277: change exponent to superscript. Done.

L278-280: We changed the units of the production rate to "atoms $(g\ quartz)^{-1}\ yr^{-1}$. Note: It is also changed in Table 2!

L280: We replaced "stopped" with "slow".

L281: We changed the notation accordingly.

L282: We clarified that the CRONUScalc online calculator has been used.

L284-285: format length and density units. Done.

L287: please move to the 'Results' section. We deleted it. It is redundant with other statements.

**Results**

L293: the three units should be mentioned now. Done.

L357: I recommend to move it to the discussion section. Done with some rephrasing: In this sense, the two upper radionuclide ages (> 2 Ma BP) are reversals (Fig. 4), which can be explained by reworking of old material that has been eroded and transported from the catchment prior to its final deposition in the Ejina Basin. In contrast, the lower sample age at 53.1 m depth (0.84 ±0.12 Ma BP) is accepted as supporting the palaeomagnetic depth-to-age-distribution, because of its approximate overlap with it.

L363, L366, L367: change to "cm ka$^{-1}$". Done.

L389: move it to the discussion section. Done.

L421: Done; moved to discussion chapter.

L430-431: Done; moved to discussion chapter.

L437-438: Done; moved to discussion chapter.

L439: Done; decimal separators corrected.

L445-447: Done; moved to discussion chapter.

L450-458: Done; shortened and moved to discussion chapter.

L526: Done; typo deleted, *Chenopodiaceae* in italics.

L556, L557: Done; change to "ka$^{-1}$".

L616-618: Further support is now given for the reasoning of Heihe fan as an upwind source to the loess plateau. The statement is backed-up with an additional reference (Herzschuh et al., 2019).

L627: abbreviation MPT is not introduced. In fact, MPT is introduced; see L621.

L629: Done; subscript in CaCO$_3$.